# Snap29 Is Dispensable for Self-Renewal Maintenance but Required for Proper Differentiation of Mouse Embryonic Stem Cells

**DOI:** 10.3390/ijms24010750

**Published:** 2023-01-01

**Authors:** Yumei Jia, Zhaoyuan Guo, Jiahao Zhu, Guanyu Qin, Wenwen Sun, Yu Yin, Haiying Wang, Renpeng Guo

**Affiliations:** 1College of Food Science and Technology, Nanjing Agricultural University, Nanjing 210095, China; 2Yunnan Key Laboratory of Primate Biomedical Research, Institute of Primate Translational Medicine, Kunming University of Science and Technology, Kunming 650500, China; 3State Key Laboratory of Medicinal Chemical Biology, Department of Cell Biology and Genetics, College of Life Sciences, Nankai University, Tianjin 300071, China

**Keywords:** ESCs, Snap29, pluripotency, autophagy

## Abstract

Pluripotent embryonic stem cells (ESCs) can self-renew indefinitely and are able to differentiate into all three embryonic germ layers. Synaptosomal-associated protein 29 (Snap29) is implicated in numerous intracellular membrane trafficking pathways, including autophagy, which is involved in the maintenance of ESC pluripotency. However, the function of Snap29 in the self-renewal and differentiation of ESCs remains elusive. Here, we show that *Snap29* depletion via CRISPR/Cas does not impair the self-renewal and expression of pluripotency-associated factors in mouse ESCs. However, *Snap29* deficiency enhances the differentiation of ESCs into cardiomyocytes, as indicated by heart-like beating cells. Furthermore, transcriptome analysis reveals that *Snap29* depletion significantly decreased the expression of numerous genes required for germ layer differentiation. Interestingly, *Snap29* deficiency does not cause autophagy blockage in ESCs, which might be rescued by the SNAP family member Snap47. Our data show that Snap29 is dispensable for self-renewal maintenance, but required for the proper differentiation of mouse ESCs.

## 1. Introduction

Embryonic stem cells (ESCs) are pluripotent cells that can self-renew indefinitely under optimal culture conditions and retain the potential to generate any somatic lineage as well as germ cells [1]. Thus, ESCs are promising donor cell sources for regenerative medicine and cellular agriculture. The pluripotency of ESCs is mainly maintained by feed-forward networks formed by the transcription factors Oct4, Sox2, and Nanog [2,3]. Quantitative balances of these factors are necessary for the self-renewal and differentiation of ESCs [4]. Using a standard test of the embryoid body (EB) formation, ESCs are able to spontaneously differentiate into three embryonic germ layers: the ectoderm, mesoderm, and endoderm [5]. Hence, ESCs are regarded as an appropriate in vitro model for investigating gene regulation, signal transduction, and embryogenesis [6,7].

Snap29 is a soluble *N*-ethylmaleimide-sensitive factor (NSF) attachment protein receptor (SNARE) that mediates membrane fusion of vesicles and target membranes in eukaryotic cells [8]. In mammals, the *SNAP29* gene is expressed in many tissues and is involved in various membrane trafficking processes, including synaptic transmission [9], endocytic recycling [10], and autophagy [11,12,13]. Autophagy is a highly conserved catabolic process with well-organized membrane dynamics [14]. In this process, the isolation membrane of the phagophore first recruits multiple proteins and expands to form a double-membrane autophagosome, which fuses with lysosomes to generate autolysosomes, where sequestrated materials are degraded [14]. As identified previously, Snap29 acts as a Qbc-SNARE and promotes autophagosome–lysosome fusion by interacting with the Qa-SANRE Stx17 and R-SNARE Vamp7/8 [11,15] or separately forming another SNARE complex with Ykt6 and Stx7 [12,16]. Based on the above findings, Snap29 is indispensable for SNARE complex formation and the fusion of autophagosomes and lysosomes. Furthermore, autophagic homeostasis is required for the maintenance of ESC pluripotency [17,18]. However, the role of Snap29 in the self-renewal and differentiation of ESCs remains elusive.

In this study, the *Snap29* gene was depleted via CRISPR/Cas9 in mouse ESCs, and the self-renewal capacity, differentiation potential, and autophagic flux were measured in *Snap29^+/+^* and *Snap29^−/−^* ESCs. Our data reveal that the self-renewing capacity of *Snap29^−/−^* ESCs is well-maintained, while the differentiation potential is severely impaired in ESCs with *Snap29* deficiency. 

## 2. Results

### 2.1. Snap29 Deficiency Does Not Impair the Self-Renewal of ESCs

To explore the function of Snap29 in ESCs, we attempted to knock out *Snap29* with the CRISPR/Cas9 system. As exon 1 of the mouse *Snap29* gene partially overlaps with the *Pi4ka* gene (Ensemble database), Cas9 sgRNA targeting exon 2 of *Snap29* was designed (Appendix A). Next, 2 knockout clones separately with a 28 bp deletion (KO-1) and a 1 bp insertion (KO-2) were obtained (Appendix A), and the disruption of the Snap29 protein was confirmed using Western blot analysis (Appendix A). Then, we investigated the self-renewal ability of *Snap29^+/+^* and *Snap29^−/−^* ESCs. *Snap29*-deficient ESCs formed compact cell clones with distinct boundaries and high alkaline phosphatase activity, which were comparable to those of wild-type ESCs (Figure 1A). Interestingly, *Snap29* depletion promoted the proliferative capacity of ESCs (Figure 1B and Appendix A). Although the expression levels of *Klf4* and *Esrrb* were moderately increased in *Snap29^−/−^* KO-2 cells through quantitative real-time PCR (qPCR), we did not observe significant expression changes in the core pluripotent markers *Oct4*, *Nanog* and *Sox2* between *Snap29^+/+^* and *Snap29^−/−^* ESCs (Figure 1C). Western blot analysis and immunofluorescence staining confirmed the retained expression of pluripotency-associated factors upon *Snap29* deficiency (Figure 1E–H). Furthermore, representative lineage marker genes were generally not activated in *Snap29^−/−^* ESCs (Figure 1D). 

To exclude potential off-target effects of the CRISPR/Cas system, *Snap29* was silenced in ESCs using two separate short hairpin RNAs (shRNAs). Consistently, *Snap29* knockdown did not alter clonal patterns, alkaline phosphatase activity, proliferation, or the expression levels of core pluripotent markers of ESCs (Appendix A).

Next, we performed RNA sequencing (RNA-seq) to analyze the systemic effects of *Snap29* depletion on ESCs. Because they displayed more differences than WT ESCs (Figure 1 and Appendix A), KO-2 cells were selected for transcriptome analysis. Compared with *Snap29^+/+^* ESCs, 424 genes were upregulated and 514 genes were downregulated in *Snap29^−/−^* ESCs (Figure 2A, *P*adj < 0.05, fold change ≥ 2). Consistent with the qPCR results, the expression levels of the pluripotency genes *Oct4*, *Sox2*, *Klf4,* and *Esrrb* were similar between *Snap29^+/+^* and *Snap29^−/−^* ESCs (Figure 2B). We took advantage of previously reported lists of genes required for pluripotency maintenance [6], and found no significantly altered pluripotency scores between *Snap29^+/+^* and *Snap29^−/−^* ESCs (Figure 2C). Meanwhile, representative lineage marker genes were expressed at quite low levels overall (based on low FPKM < 5.0) in ESCs regardless of whether they had a *Snap29* deficiency or not (Figure 2D). We then performed Gene Ontology (GO) analysis using differentially expressed genes (DEGs) and found numerous membrane-related cellular components (Figure 2E), which is consistent with the membrane trafficking role of Snap29 [8,10]. Furthermore, DEGs were enriched in several pathways, including breast cancer, basal cell carcinoma, transcription misregulation in cancer, gastric cancer, and others (Figure 2F). 

Taken together, we successfully established the *Snap29* knockout ESC lines and found that the disruption of *Snap29* did not influence the self-renewal capacity or the maintenance of pluripotent markers of ESCs. 

### 2.2. Snap29 Depletion Promotes ESC Differentiation into Heart-like Beating Cells

To examine whether Snap29 is essential for the differentiation of mouse ESCs, we performed *in vitro* differentiation using a standard embryoid body (EB) formation assay. *Snap29^+/+^* and *Snap29^−/−^* ESCs could form EBs with smooth and round edges, while EBs formed with *Snap29^−/−^* cells were significantly larger than those of the control group (Figure 3A). Upon differentiation, all ESCs showed a significantly reduced expression of pluripotency markers, including Oct4, Nanog and Sox2 (Appendix A), indicating that the repression of pluripotency factors is not affected by *Snap29* knockout. During differentiation, heart-like beating cells emerged from the EB-derived cell cultures. Surprisingly, while approximately 60% of EBs formed from *Snap29^−/−^* ESCs exhibited heart-like beating, only 42% of the control cells showed visible beating (Figure 3C). The beating frequency (times per minute) was similar between *Snap29^+/+^* and *Snap29^−/−^* ESC-derived cells (Figure 3D). Moreover, differentiation of *Snap29^+/+^* and *Snap29^−/−^* ESCs both yielded cells representing three embryonic germ layers, as indicated by the tissue-specific immunofluorescence staining of βIII-tubulin (neurons, ectoderm), AFP (liver, endoderm), and α-SMA (smooth muscle, mesoderm) (Figure 3E–G). Nevertheless, *Snap29* depletion improved the expression of α-SMA in differentiated cells (Figure 3G,H), suggesting the enhanced mesoderm specification of ESCs without *Snap29*.

### 2.3. Snap29 Deficiency Leads to Defective Differentiation of ESCs

To further reveal the role of Snap29 in the differentiation of mouse ESCs, we performed RNA-seq analysis using *Snap29^+/+^* and *Snap29^−/−^* ESC-derived differentiated cells at day 14. Notably, 291 genes were upregulated and 545 genes were downregulated in *Snap29^−/−^* cells when compared with *Snap29^+/+^* cells (Figure 4A, *P*adj < 0.05, fold change ≥ 2). Interestingly, numerous myogenesis-related terms were top-ranked in GO analysis by all DEGs, including myotube differentiation, cardiac muscle contraction, regulation of myotube differentiation, striated muscle contraction in Biological Process, contractile fiber, striated muscle thin filament, myofibril, myofilament, contractile fiber part in Cellular Component, extracellular matrix structural constituent, and tropomyosin binding in Molecular Function (Figure 4B). We then measured the FPKMs of GO-enriched genes and found that the majority of them were significantly upregulated in *Snap29^−/−^* cells, and representative genes are shown in Figure 4C. Consistently, the expression levels of the genes with the highest FPKM values, including *Acta1*, *Col5a2*, *Jam2*, *Cxcl10,* and *Trim54*, were significantly increased in *Snap29^−/−^* cells through qPCR (Figure 4D). These results indicate that *Snap29* deficiency significantly promotes the expression of genes associated with muscle lineage differentiation and formation, which is consistent with the increased incidence of heart-like cells upon *Snap29* depletion (Figure 3). Moreover, adhesion molecules, cytokine receptor interaction, focal adhesion, PI3K-Akt signaling pathways, and ECM-receptor interaction were included in enriched pathways by KEGG analysis (Figure 4E), and might be involved in the regulation of myogenesis. 

Next, we investigated the functional role of 545 genes downregulated in *Snap29^−/−^* cells (Figure 4A). Taking advantage of previously reported lists of the genes essential for the germ layer differentiation of pluripotent stem cells (PSCs) [19], we found that 23 downregulated genes were required for the definitive endoderm differentiation of PSCs, including *Ccnd2*, *Hesx1*, *Flt1,* and *Lonrf3*; 8 genes for early mesoderm formation, including *Tubb4A*, *Kif21B*, *Cdx2,* and *Cacna1H*; and 16 genes for neuroectoderm differentiation, including *Snap29* itself, *Fstl3*, *Thyh3,* and *II3Ra* (Figure 5A). The FPKMs of related genes are shown in Figure 5B and verified using a qPCR assay (Figure 5C). In GO analysis, embryonic organ development was top-ranked in the enriched biological process using downregulated genes upon *Snap29* disruption (Figure 5D). Consistently, pathways regulating cell differentiation and organ development, including the PI3K-Akt pathway, Ras signaling pathway, and MAPK signaling pathway, were enriched in KEGG pathways (Figure 5E). Together, transcriptome analysis strongly suggests that *Snap29* deficiency significantly disturbs the expression of the genes related to embryonic germ layer differentiation and further embryonic organ development. 

### 2.4. Snap29 Depletion Does Not Cause Autophagic Blockage in ESCs and Differentiated Cells

Next, the role of the autophagy modulation of Snap29 in ESCs was measured. In contrast to previous studies in nonembryonic cells [11,15], *Snap29* knockout or knockdown in mouse ESCs did not lead to the accumulation of the autophagosome marker LC3-II or the universal autophagic substrate SQSTM1/p62 (Figure 6A and Appendix A). Upon differentiation, less LC3-II turnover and retained p62 protein were observed in *Snap29* knockout cells at day 7 (Figure 6B). However, at day 14 of differentiation, the LC3-II/LC3-I ratio was increased in *Snap29^−/−^* cells in comparison with *Snap29^+/+^* cells (Figure 6C). These results suggest that Snap29 is not necessarily required for autophagosome-lysosome fusion in mouse ESCs. Given the essential role of the SNARE complex in autophagy regulation, we speculate that other SNAP proteins might compensate in *Snap29*-deficient ESCs. The SNAP family contains four known members in mammals: Snap23, Snap25, Snap29, and Snap47, all of which contain two separate SNARE motifs [8]. Based on our RNA-seq data, we found that the expression of *Snap25* was extremely low in mouse ESCs and differentiated cells (FPKM < 0.1) and that the expression of *Snap47* was upregulated in *Snap29* knockout cells (Figure 6D). Then, we knocked down *Snap47* and *Snap23* using small interfering RNAs (siRNAs) in *Snap29^+/+^* and *Snap29^−/−^* ESCs (Figure 6E). Interestingly, *Snap47* knockdown, but not *Snap23* knockdown, resulted in the obvious accumulation of LC3-II and p62 in the *Snap29* knockout ESCs (Figure 6F), indicating autophagic blockage in ESCs with dual disruption of *Snap29* and *Snap47*. Taken together, these data suggest that *Snap29* knockout does not lead to defective autophagic flux in ESCs and differentiated cells, which might be due to the rescue effect mediated by paralogous *Snap47*. 

## 3. Discussion

Snap29 plays a role in membrane trafficking and autophagy modulation [8,13]. Here we show that in pluripotent mouse ESCs, *Snap29* knockout or knockdown does not impair cell proliferation, colony formation, alkaline phosphatase activity, or the expression of pluripotent markers (Figure 1 and Appendix A). Furthermore, transcriptome analysis confirmed that the core pluripotent factors are well-maintained in *Snap29^−/−^* ESCs (Figure 2). Interestingly, the DEGs between *Snap29^+/+^* and *Snap29^−/−^* ESCs were enriched in several cancer-related pathways (Figure 2F). The evidence demonstrates that the altered Snap29 activity might contribute to the pathogenesis of cancers [20]. Recent evidence from a breast cancer cell line and a human melanoma cell line suggests that Snap29 plays a role in the TNFα-NF-κB-FOXP3 signaling axis, and silencing *Snap29* promotes tumor cell migration [21]. Thus, Snap29 might represent a novel safeguard to counteract the pathogenic processes involved in cancer.

As a Qbc SNARE, the function of Snap29 in promoting autophagosome-lysosome fusion is well-established. Unexpectedly, unlike many other studies [11,13,15,22], Snap29 depletion in mouse ESCs did not lead to defective autophagosome-lysosome fusion or the accumulation of autophagosomes (Figure 6). We speculate that this unusual outcome might be caused by the O-linked β-N-acetylglucosamine (O-GlcNAc) modification of Snap29 in mouse ESCs. Guo et al. demonstrated that O-GlcNAcylated Snap29 inhibits the formation of a Snap29-containing SNARE complex, decreases the fusion of autophagosomes and endosomes/lysosomes, and blocks autophagic flux [13]. UDP-GlcNAc is a key substrate for the O-GlcNAcylation of proteins, and is produced primarily through glycolysis and hexosamine biosynthetic pathways [23]. As has been widely studied, PSCs, such as mouse ESCs, prefer glycolysis over oxidative phosphorylation to increase glucose uptake and energy supply [24]. Supplementation with glutamine in cell culture media (see Methods section) may also increase the production of UDP-GlcNAc [25]. O-GlcNAc modification is balanced by two enzymes: O-GlcNAc transferase (OGT) adds the modification and O-GlcNAcase (OGA) removes it [26]. As reported, mouse ESCs exhibit high expression of OGT, and Oct4 and Sox2 are modified with O-GlcNAc in ESCs, which is required for the activity of the pluripotency network [27]. Taken together, it is reasonable to speculate that Snap29 is O-GlcNAcylated and fails to form the SNARE complex with the associated factors in mouse ESCs. Further investigations should be performed to verify the O-GlcNAcylation of Snap29 and its biological functions in pluripotent ESCs.

In light of the fundamental role of Snap29-containing SNARE complexes in membrane trafficking, unknown Qbc SNARE domain-containing proteins may exist in mouse ESCs. In this study, preliminary findings suggest that Snap47 may compensate for Snap29 to promote autophagosome and lysosome fusion (Figure 6). Snap47 protein contains two separate SNARE domains and its silencing in HeLa cells leads to impaired autophagic flux [28]. Moreover, Snap47 can interact with ATG14 to promote viral capsid protein conjugation and coxsackievirus propagation, while its role in autophagy is undetermined [29]. Overall, the function of Snap47 has not been extensively investigated. It would be intriguing to examine the role and mechanism of Snap47 in SNARE complex formation, autophagy regulation, and pluripotency maintenance in ESCs and other types of cells. 

In this study, we performed standard *in vitro* differentiation of *Snap29^+/+^* and *Snap29^−/−^* ESCs by EB formation. This method has been extensively used to investigate regulatory factors or signaling pathways that mediate ESC differentiation toward numerous cell lineages [30,31]. Moreover, the expression of tissue-specific genes of ESCs during *in vitro* differentiation recapitulates the early process of *in vivo* development [32]. Interestingly, *Snap29^−/−^* ESCs formed significantly larger EBs than *Snap29^+/+^* cells (Figure 3A,B). Meanwhile, *Snap29* deficiency significantly promoted the generation of heart-beating cardiac cells (Figure 3). The size of EBs has been demonstrated to influence the differentiation potential of ESCs in this system [33], and larger EBs are associated with the enhanced cardiogenesis of ESCs [34].

Furthermore, the transcriptome analysis confirmed that the differentiation of ESCs into muscle lineage cells was enhanced by *Snap29* knockout (Figure 4). We then examined the expression of the genes related to myogenesis, and found that the majority of them were significantly upregulated in *Snap29^−/−^* cells (Figure 4). Representatively, Myod1 and Cav3 are essential transcription factors characterizing myogenic precursors; the *Acta1* gene encodes Actin-alpha 1, a skeletal muscle protein, and the *Tnnt* gene encodes the Troponin T (TnT) protein, both of which are fundamental proteins found in muscle tissues; the *Col4a* and *Col5a* genes encode several types of collagens, which are flexible proteins that are important in the structure of many tissues, including muscles [35]. Our data uncover the role of Snap29 in muscle lineage differentiation and myofiber formation. Nonetheless, the causal link between Snap29, myogenic factors, and myogenesis requires further investigation. 

Additionally, numerous genes that are required for germ layer differentiation are significantly misregulated in *Snap29^−/−^* cells compared to control cells, especially the genes for definitive endoderm and neuroectoderm (Figure 5). AFP is extensively used as a marker indicating the endoderm differentiation of PSCs. However, AFP^+^ cells emerged from both the *Snap29^+/+^* and *Snap29^−/−^* ESC-derived cell populations (Figure 3 F,H). The disagreement can be explained by the versatile expression of AFP, which is secreted by the primitive and definitive endoderm lineages, including the visceral endoderm, its derivative yolk sac endoderm, fetal liver hepatocytes, and the gut epithelium of the developing embryos [36]. Taken together, we assume that, during *in vitro* differentiation, although some AFP^+^ cells emerge without *Snap29*, the ability of *Snap29^−/−^* cells to differentiate into functional progenitors or somatic cells in the endoderm might be insufficient. 

ESCs are an adequate *in vitro* model for studying embryogenesis [6,7]. In humans, the loss of functional Snap29 results in CEDNIK syndrome (cerebral dysgenesis, neuropathy, ichthyosis, and keratoderma), a neurocutaneous disease characterized by severe developmental abnormalities of the nervous system and aberrant differentiation of the epidermis [37]. In addition, CEDNIK patients show signs of microcephaly and facial dysmorphism, hypoplastic optic disk, sensorineural deafness, and severe cachexia [37], indicating multisystem defects caused by *Snap29* deficiency. In this study, we reveal that numerous genes that are required for neuroectoderm differentiation are significantly downregulated upon *Snap29* knockout, including *Fstl3*, *Ttyh3,* and *Il3Ra* (Figure 5). In the immunoassays, the expression of βIII-tubulin in *Snap29^−/−^* cells was comparable to that in control cells (Figure 3). As reported previously, detecting only the expression of βIII-tubulin is inadequate to measure the neuroectoderm differentiation of PSCs [38]. In addition to βIII-tubulin and the DEGs analyzed in RNA-seq, other essential regulatory factors and signaling pathways need to be tested in the following studies. Furthermore, although several animal models have been established to recapitulate CEDNIK syndrome [39,40,41], cell models are rather scarce [42]. Our ESC model lacking *Snap29* may serve as a powerful tool for investigating the embryonic differentiation process of CEDNIK syndrome and seeking compounds that might ameliorate traits of the disease. 

In this study, we uncovered that Snap29 is not required for the self-renewal maintenance of mouse ESCs. *Snap29* depletion does not impair cell proliferation or the expression of pluripotency-associated markers in ESCs. However, Snap29 is indispensable for the proper differentiation of mouse ESCs. *Snap29* deficiency enhances the differentiation of ESCs into cardiomyocytes, as indicated by heart-like beating cells. Moreover, transcriptome analysis revealed that *Snap29* depletion significantly decreased numerous genes required for embryonic germ layer differentiation. Furthermore, *Snap29* deficiency does not cause autophagy blockage in ESCs, which might be rescued by the SNAP family member Snap47. Our study reveals the role of Snap29 in the maintenance of ESC pluripotency and provides insights for further investigation of the complicated pathogenic mechanisms of CEDNIK syndrome caused by *Snap29* deficiency.

## 4. Materials and Methods

### 4.1. Cell Culture

The mouse J1 ES cell line was kindly gifted from Pro. Lin Liu of Nankai University, which was 129S4/SvJae in origin and cultured without feeder layers [43]. For routine culture, ESCs were maintained under 5% CO_2_ at 37 °C in ESC culture medium consisting of knockout DMEM supplemented with 20% fetal bovine serum (FBS, ES quality, HyClone, Rd. Logan, UT, USA), 1000 U/mL leukemia inhibitory factor (LIF) (ESGRO, Merck Millipore, Burlington, MA, US), 0.1 mM nonessential amino acids, 0.1 mM β-mercaptoethanol, 2 mM L-glutamine, and penicillin (100 U/mL) and streptomycin (100 μg/mL). 

### 4.2. Generation of Snap29 Knockout ESCs

Snap29 knockout ESCs were generated using the CRISPR/Cas9 system. Single-guide RNAs (sgRNAs) targeting exon 2 of the mouse *Snap29* gene were designed using the online design tool available at http://crispr.genome-engineering.org/ (accessed on 28 December 2020). SgRNAs were cloned into the pSpCas9(BB)-2A-Puro (PX459) vector, and transfected into ESCs with lipofectamine^TM^ transfection reagent (Invitrogen), according to the manufacturer’s recommendation. Then, knockout ESC clones were identified through Sanger sequencing and confirmed using western blot analysis. 

### 4.3. Cell Proliferation Analysis by Accumulative Growth Curve

2 × 10^5^ WT, *Snap29* KO-1, and KO-2 ESCs were initially seeded into 6-wells and cultured for 2 days. Then, ESCs were digested using Typsin-EDTA, counted by the hemocytometers, and passaged at certain splitting ratios of 1:5–1:12. ESCs were passaged for 6 days. The accumulative growth curve was drawn using the product of the cell numbers and the splitting ratios. 

### 4.4. MTT Assay and AP Staining

The methyl thiazolyl tetrazolium (MTT) assay was used to measure cell viability, and committed using an MTT Cell Proliferation and Cytotoxicity Assay Kit (Beyotime), according to the manufacturer’s instructions. Alkaline phosphatase (AP) staining was performed with a BCIP/NBT Alkaline Phosphatase Colour Development Kit (Beyotime), according to the manufacturer’s instructions.

### 4.5. In Vitro Differentiation of ESCs

For ESC differentiation, three-dimensional colonies known as embryoid bodies (EBs) were generated [30]. Briefly, undifferentiated ESCs were trypsinized to obtain a single cell suspension, and EBs were formed in ESC culture medium without LIF, in a definite number of cells in hanging drops for 3 days. Then, EBs were transferred to 24-well culture plates with 1 EB per well. Daily microscopic observations were conducted to detect beating EBs. In total, 20 to 30 EBs were transferred to 6-well microwell plates per well for protein, RNA, and sample collection.

### 4.6. Gene Expression by Quantitative Real-Time PCR

Total RNA was isolated from samples using the MiniBEST Universal RNA Extraction Kit (Takara), according to the manufacturer’s protocol. The purity and concentration of RNA were checked using Nanodrop technology. Then, 1 μg RNA was subjected to cDNA synthesis using PrimeScript^TM^ RT Master Mix (Takara). Quantitative real-time PCR (qPCR) reactions were set up in duplicate with the ChamQ SYBR qPCR Master Mix (Vazyme) and run on the QuantStudio^TM^ 5 Real-time PCR Instrument (Applied Biosystems). Each sample was repeated at least twice and analyzed with GAPDH serving as the internal control. Quantification of gene expression was based on the Ct (cycle threshold) value. Melting curve analysis was performed to control PCR product specificities and exclude nonspecific amplification. Primers for qPCR are listed in Appendix A.

### 4.7. Western Blot

Cells were collected and resuspended in NP40 cell lysis buffer containing 1 mM PMSF (Beyotime) and protease inhibitor cocktail (Bimake). Twenty micrograms of protein was separated on SDS-polyacrylamide gels and transferred to polyvinylidene difluoride (PVDF) membranes. Membranes were blocked with 5% nonfat milk or 5% BSA in 1× TBST at room temperature for 2 h. Samples were probed with primary antibodies overnight at 4 °C (for antibody details, see Appendix A). HRP-conjugated goat anti-rabbit IgG or goat anti-mouse IgG secondary antibodies were diluted at 1:5000. Protein bands were detected using Super Sensitive ECL Luminescent reagents (Qinxiang). The band intensity was measured using ImageJ software and normalized to the intensity of β-tubulin. The relative expression level was calculated from the results of at least three independent experiments or samples.

### 4.8. Immunofluorescence

ESCs were grown on gelatin-treated coverslips and washed twice in PBS, then fixed in freshly prepared 4% paraformaldehyde for overnight at 4 °C, permeabilized in 0.1% Triton X-100 in blocking solution (3% goat serum in PBS) for 30 min, washed three times, and left in blocking solution for 1.5 h. Samples were incubated overnight at 4 °C with primary antibodies (for antibody details, see Appendix A), washed three times, and incubated for two hours with the secondary antibodies, goat anti-mouse IgG (H +L) FITC or goat anti-rabbit IgG (H + L) Alexa Fluor 594 (Jackson). Samples were washed twice with PBS, stained with 0.5 μg/mL DAPI, and mounted in Vectashield mounting medium. Fluorescence was detected and imaged using a Leica Microsystems CMS microscope.

### 4.9. RNA Sequencing and Bioinformatics Analysis

ESCs or differentiated cells were harvested and total RNA was extracted using TRIzol reagent and according to the manufacturer’s instructions. The quality control method was mainly performed using an Agilent 2100 bioanalyzer. The initial RNA for library construction was total RNA, and the mRNA with a poly A tail was enriched with oligo (dT) magnetic beads. Using fragmented mRNA as a template and random oligonucleotides as primers, the first strand of cDNA was synthesized in the M-MuLV reverse transcriptase system, which was followed by second strand cDNA synthesis. The cDNA fragments were added to a single “A” base and subsequently ligated with the adapter. The products were then purified and enriched with PCR amplification. The PCR yield was quantified and samples were pooled together to make a single-stranded DNA circle (ssDNA circle), generating the final library. The clustering of the index-coded samples was performed on a cBot Cluster Generation System using TruSeq PE Cluster Kit v3-cBot-HS (Illumina) according to the manufacturer’s instructions. After cluster generation, the library preparations were sequenced on an Illumina NovaSeq platform and 150 bp paired-end reads were generated.

For bioinformatics analysis, the clean reads were mapped to the Mus musculus mm10 reference genome using HISAT2 v2.0.5. Gene quantification (read counts) was calculated by FeatureCounts (1.5.0-p3). Then the normalized gene expression (FPKM) was calculated with read counts and gene lengths. DESeq2 was used to identify differentially expressed genes. Functional enrichment (GO annotation, KEGG) of gene sets with different expression patterns was performed using cluster Profiler.

### 4.10. Knockdown by shRNA or siRNA

For shRNA-mediated knockdown, interfering oligonucleotides were synthesized and cloned into the pSIREN-RetroQ vector, according to the manufacturer’s instructions. The packaging cell line PT67 was transfected with 2 μg plasmid using lipofectamine^TM^ transfection reagent (Invitrogen). After 48 h, the supernatant of the recombinant retrovirus was collected and used to infect ESCs. The clones were selected by 2 μg/mL puromycin, and surviving ones were picked. Subsequently, the knockdown efficiency was identified using qPCR. For siRNA-mediated knockdown, synthesized oligonucleotides were transfected into ESCs with lipofectamine^TM^ transfection reagent (Invitrogen). Forty-eight hours later, ESCs were collected and analyzed for qPCR or western blotting. Sequences of oligonucleotides for shRNA and siRNA are listed in Appendix A.

### 4.11. Statistical Analysis

Data were analyzed by two-tailed Student’s *t*-test or χ^2^ test depending on specific experiments, and the *p* value was calculated. Statistical significance was defined as * *p* < 0.05, ** *p* < 0.01 or *** *p* < 0.001.

## Figures and Tables

**Figure 1 ijms-24-00750-f001:**
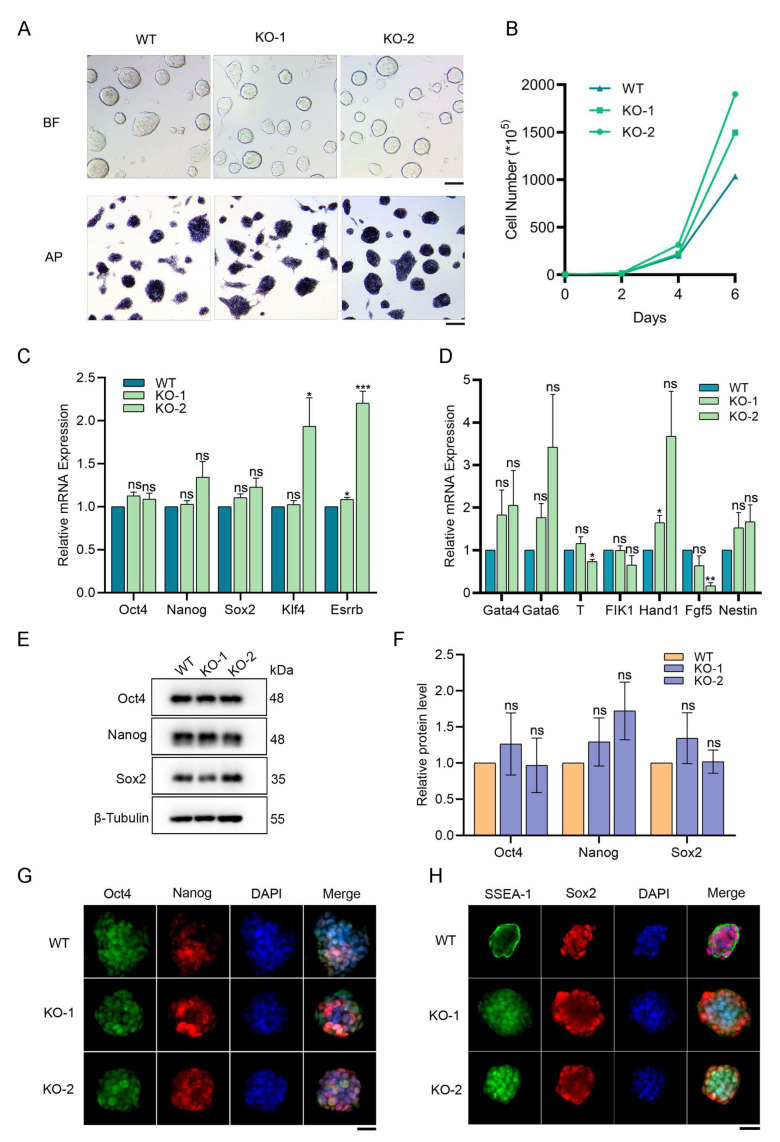
The self-renewal ability of ESCs is maintained upon *Snap29* depletion. (**A**) Morphology of *Snap29*^+/+^ (WT) and *Snap29*^−/−^ (KO) ESCs under bright field (BF) and alkaline phosphatase (AP) staining analysis. KO-1: -28 bp; KO-2, +1 bp. Scale bar, 100 μm. (**B**) An accumulative growth curve of *Snap29*^+/+^ and *Snap29*^−/−^ ESCs. (**C**) Expression analysis of pluripotent marker genes in *Snap29*^+/+^ and *Snap29*^−/−^ ESCs by qPCR. (**D**) Expression analysis of lineage marker genes in *Snap29*^+/+^ and *Snap29*^−/−^ ESCs by qPCR. (**E**) Western blot analysis of Oct4, Nanog, and Sox2 in ESCs. (**F**) Quantity using ImageJ software of pluripotent protein levels relative to β-tubulin as a loading control. (**G**,**H**) Immunofluorescence of the pluripotency-associated markers Oct4, Nanog, Sox2, and SSEA-1. Scale bar, 25 μm. Mean ± SEM from three independent experiments. * *p* < 0.05, ** *p* < 0.01, *** *p* < 0.001, ns not significant (*p* > 0.05), unpaired two-tailed Student’s *t*-test, compared with WT groups.

**Figure 2 ijms-24-00750-f002:**
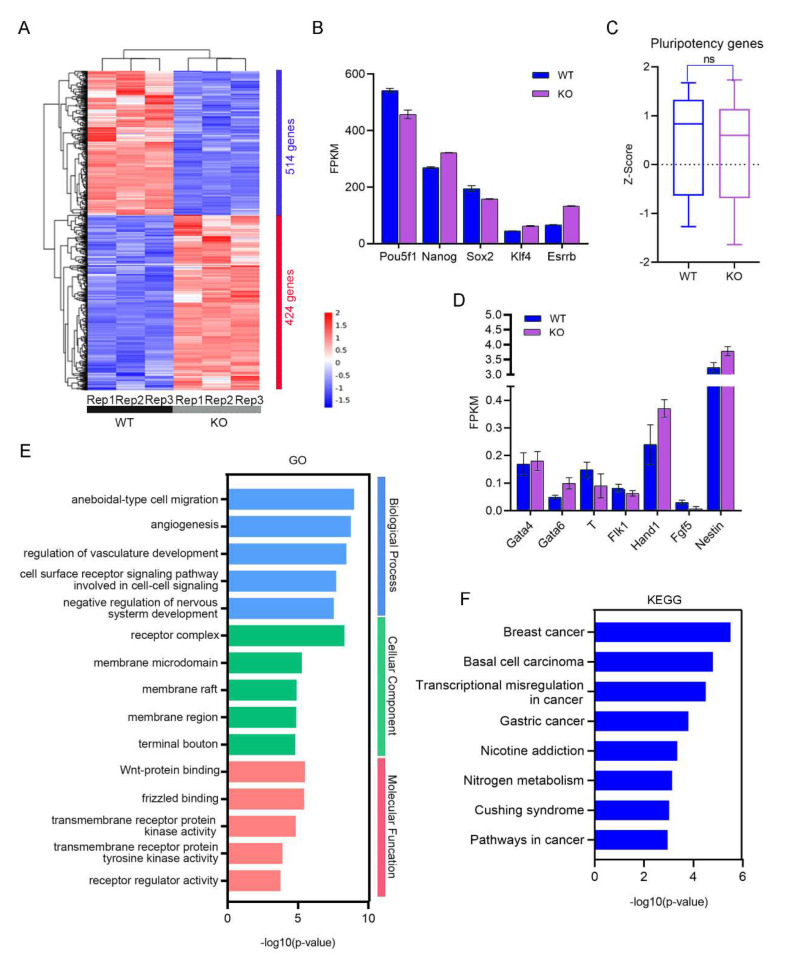
Transcriptome analysis using RNA-seq in *Snap29*^+/+^ and *Snap29*^−/−^ KO ESCs. (**A**) Heatmap illustrating differentially expressed genes (DEGs) between *Snap29*^+/+^ and *Snap29*^−/−^ ESCs. WT and KO-2 ESCs were used here, and three biological replicates were analyzed per group. Genes with ≥2-fold expression changes, and *P*adj < 0.05 were chosen for the heatmap. (**B**) Expression levels (FPKMs) of pluripotency genes in *Snap29*^+/+^ and *Snap29*^−/−^ ESCs. (**C**) Z-scores of genes that were required for pluripotency maintenance [6]. (**D**) FPKMs of lineage marker genes in *Snap29*^+/+^ and *Snap29*^−/−^ KO ESCs. (**E**) GO analysis of DEGs between *Snap29*^+/+^ and *Snap29*^−/−^ ESCs. (**F**) The enriched KEGG pathways of DEGs between *Snap29*^+/+^ and *Snap29*^−/−^ ESCs. Mean ± SEM from three biological replicates.

**Figure 3 ijms-24-00750-f003:**
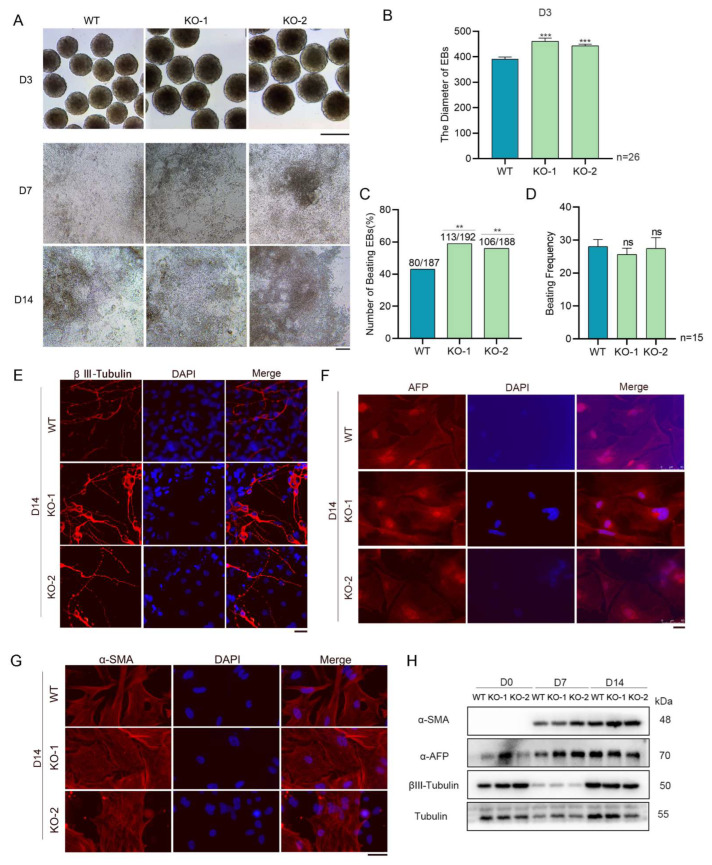
*Snap29* depletion enhances the diameters of EBs and the incidence of beating EBs. (**A**) Morphology of embryoid bodies (EBs) at day 3 (scale bar, 400 μm), and differentiated cells at day 7 and day 14 (scale bar, 50 μm) derived from *Snap29*^+/+^ and *Snap29*^−/−^ ESCs. (**B**) The diameters of day 3 EBs in (**A**). (**C**) Incidence of derivation of beating EBs in percentage (%). (**D**) Beating frequency of EBs (times per minute). (**E**–**G**) The differentiated derivatives of day 14 consist of cells representing three embryonic germ layers, as indicated by immunofluorescence staining of markers for ectoderm (βIII-Tubulin), endoderm (AFP), and mesoderm (α-SMA). Scale bar = 25 μm. (**H**) Western blot analysis of α-SMA, AFP, and βIII-tubulin in the indicated cells. Mean ± SEM from three independent experiments. ** *p* < 0.01, *** *p* < 0.001, ns not significant (*p* > 0.05), unpaired two-tailed Student’s *t*-test for (**B**) and (**D**), χ^2^ test for (**C**), compared with WT groups.

**Figure 4 ijms-24-00750-f004:**
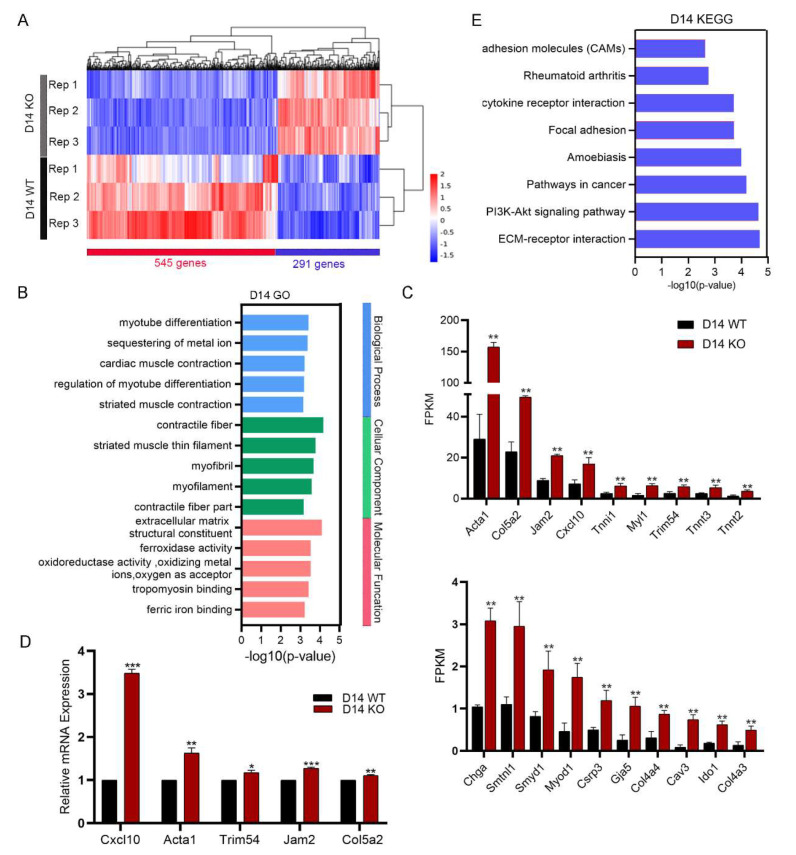
Transcriptome analysis using RNA-seq in *Snap29*^+/+^ and *Snap29*^−/−^ differentiated cells. (**A**) Heatmap illustrating DEGs between *Snap29*^+/+^ and *Snap29*^−/−^ differentiated cells at day 14. Three biological replicates were analyzed per group. Genes with ≥2-fold expression changes, and *P*adj < 0.05 were chosen for the heatmap. (**B**) The GO analysis of DEGs between *Snap29*^+/+^ and *Snap29*^−/−^ differentiated cells. (**C**) FPKMs of genes involved in the development of muscle lineage cells. (**D**) Expression analysis of representative genes in (**C**) by qPCR. (**E**) The enriched KEGG pathways of DEGs between *Snap29*^+/+^ and *Snap29*^−/−^ differentiated cells. Mean ± SEM from three biological replicates. * *p* < 0.05, ** *p* < 0.01, *** *p* < 0.001, ns not significant (*p* > 0.05), unpaired two-tailed Student’s *t*-test.

**Figure 5 ijms-24-00750-f005:**
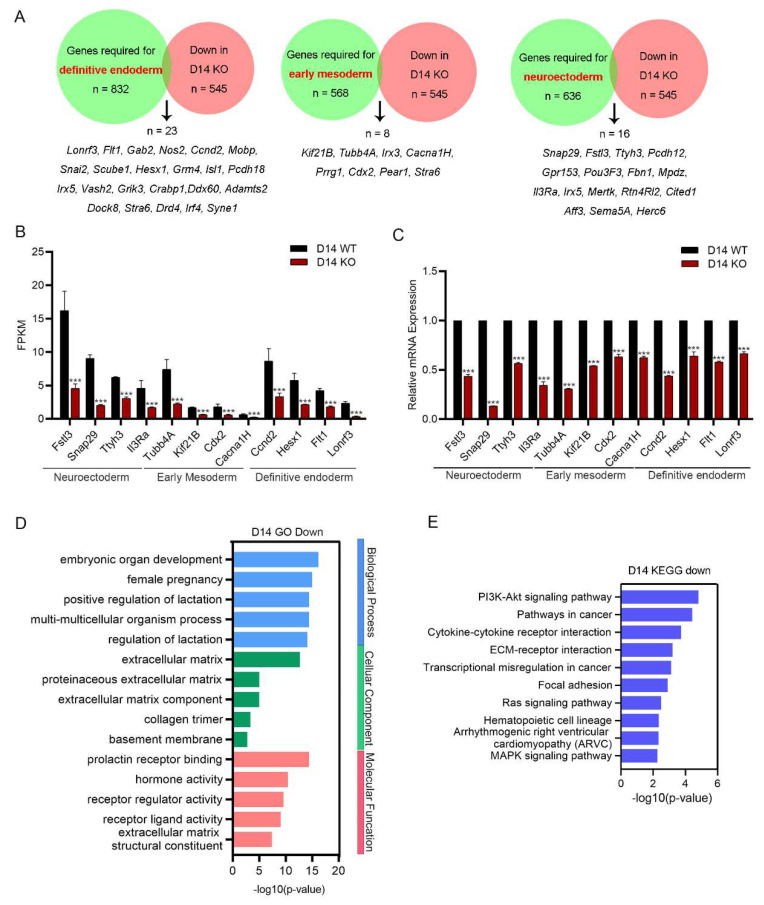
Downregulated genes in *Snap29^−/−^* differentiated cells are involved in the proper differentiation of ESCs. (**A**) Venn diagrams of downregulated genes in *Snap29*^−/−^ differentiated cells overlapping with genes required for definitive endoderm, early mesoderm, and neuroectoderm formation [19]. (**B**) FPKMs of representative genes involved in the development of the definitive endoderm, early mesoderm, and neuroectoderm. (**C**) Expression analysis of genes in (**B**) through qPCR. (**D**) GO analysis of downregulated genes in *Snap29*^−/−^ differentiated cells. (**E**) The enriched KEGG pathways analyzed through downregulated genes in *Snap29*^−/−^ differentiated cells. Mean ± SEM from three biological replicates. *** *p* < 0.001, unpaired two-tailed Student’s *t*-test.

**Figure 6 ijms-24-00750-f006:**
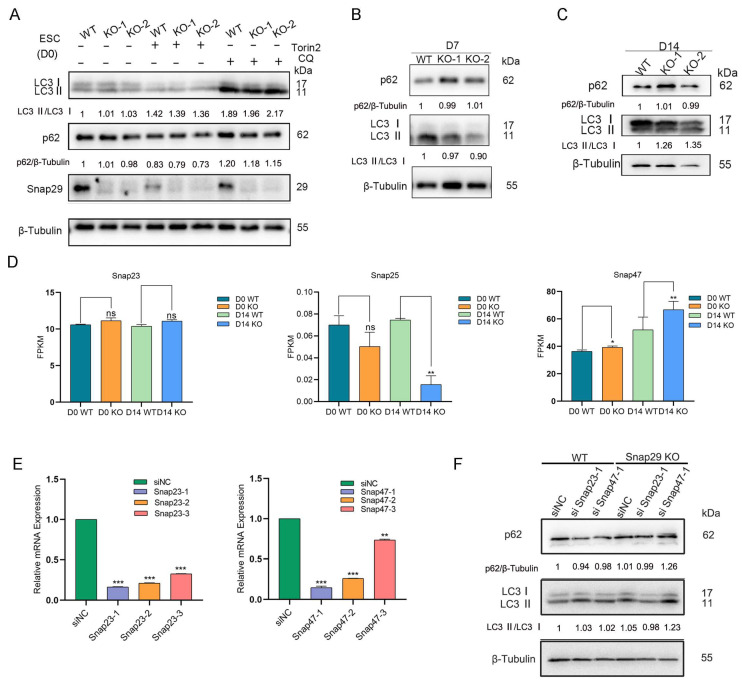
*Snap29* deficiency does not lead to impaired autophagy in ESCs and differentiated cells. (**A**) Western blot analysis of LC3 I/II and p62 levels in *Snap29*^+/+^ and *Snap29*^−/−^ ESCs. Torin2 (200 nM) and chloroquine (CQ, 50 μM) were used for 6 h to induce or block autophagic flux, respectively. (**B**) Western blot analysis of LC3 I/II and p62 levels in *Snap29*^+/+^ and *Snap29*^−/−^ differentiated cells at day 7. (**C**) Western blot analysis of LC3 I/II and p62 levels in *Snap29*^+/+^ and *Snap29*^−/−^ differentiated cells at day 14. (**D**) FPKMs of *Snap23*, *Snap25,* and *Snap47* in *Snap29*^+/+^ and *Snap29*^−/−^ ESCs (D0) and differentiated cells at day 14 (D14). (**E**) I Knockdown efficiency of *Snap23* and *Snap47* separately by three siRNAs. (**F**) Western blot analysis of LC3 I/II and p62 levels in *Snap29*^+/+^ and *Snap29*^−/−^ ESCs upon *Snap23* or *Snap47* knockdown. Quantifications of relative protein levels from the Western blot assays are indicated below the corresponding bands in A, B, C, and F. Mean ± SEM from three independent experiments. * *p* < 0.05, ** *p* < 0.01, *** *p* < 0.001, ns not significant (*p* > 0.05), unpaired two-tailed Student’s *t*-test, compared with control groups.

## Data Availability

The data and materials supporting the findings of this study are available within the article or the Appendix A. RNA-Seq data have been deposited in the National Center for Biotechnology Information Gene Expression Omnibus (GEO) database (https://www.ncbi.nlm.nih.gov/geo, accessed on 20 June 2022) under the GEO Series accession number GSE206440. The datasets used and/or analyzed during the current study are available from the corresponding author on reasonable request.

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
