# Peer review of "Snap29 Is Dispensable for Self-Renewal Maintenance but Required for Proper Differentiation of Mouse Embryonic Stem Cells"

_ijms, 2023, doi:10.3390/ijms24010750_

Round 1

Reviewer 1 Report

The manuscript raised an interesting topic in ES cell study.

Comments,

Authors claimed that Snap29 deficiency does not impair the self-renewal of ESCs, and Snap29 was knocked out with CRISPR/Cas9 system. From Fig 2B, Snap29 deficiency promotes cell proliferation; How about the efficiency of Snap29 KO by CRISPR/Cas9 system? How many passages of Snap29 KO ES cells were tested.

Surprisingly, while approximately 60% of EBs formed from Snap29-/- ESCs exhibited heart-like beating, only 42% of the control cells showed visible beating. A video should be attached, and a histology analysis should be done.  Only cardiomyocyte differentiation was tested, how about other germ layer differentiation of Snap29-/- ESCs.

Author Response

Response to Reviewer 1 Comments

Thank you very much for reviewing our manuscript. The thoughtful comments are quite helpful to us. As below, I would like to clarify the concerns point by point.

>Comments and Suggestions for Authors

The manuscript raised an interesting topic in ES cell study.

Comments,

Authors claimed that Snap29 deficiency does not impair the self-renewal of ESCs, and Snap29 was knocked out with CRISPR/Cas9 system. From Fig 2B, Snap29 deficiency promotes cell proliferation; How about the efficiency of Snap29 KO by CRISPR/Cas9 system? How many passages of Snap29 KO ES cells were tested.

Response: Thank you for the precise summary of the findings reported in this manuscript.

  • In KO experiments, we picked 29 cell clones and found that at least one allele of Snap29 gene was edited in 24 clones. Among them, 10 clones were assumed to be Snap29-/- (with indels not a multiple of 3 nt). Thus, the the efficiency of Snap29 KO by CRISPR/Cas9 system was high and sufficient to perform the following experiments.
  • We measured Snap29 KO ES cells in passage 5~6 after stable clonal establishment by Western blot analysis and validated the results in passage 12. For other assays in our study, WT and Snap29 KO ES cells at passage 4~6 were used.

Surprisingly, while approximately 60% of EBs formed from Snap29-/- ESCs exhibited heart-like beating, only 42% of the control cells showed visible beating. A video should be attached, and a histology analysis should be done.  Only cardiomyocyte differentiation was tested, how about other germ layer differentiation of Snap29-/- ESCs.

Response: Thank you for the constructive comments. 

  • Representative videos have been included in the Supplemental Video 1-3 (WT, KO-1,KO-2) Nevertheless, due to the COVID pandemic, histology analysis can’t be performed in recent days. To further support our views, we moved immunofluorescence images of three germ layer markers (initially Fig. S3 C-E) to the main manuscript (revised Figure 3, revised text Line 267-273). Meanwhile, we added the relevant Western blot results detecting α-SMA, AFP, and βIII-tubulin (revised Figure 3, revised text Line 271-273). Our data indicate that the molecular marker of mesoderm α-SMA, is upregulated in Snap29 KO differentiated cells in comparison with control cells. The results are consistent with our main findings obtained by heart-like beating and transcriptome analysis.
  • For ectoderm and endoderm differentiation analysis, we detected expression levels of βIII-tubulin and AFP, and found no significant changes in Snap29 KO cells (revised Figure 3, revised text Line 263-273). Furthermore, in RNA-seq analysis, we found 16 genes required for nueroectoderm formation and 23 genes for definitive endoderm differentiation were significantly downregulated, both greater than genes essential for early mesoderm formation (8 genes, revised text 307-314). Taken together, we summarize that, although maintaining the ability of forming βIII-tubulin+ cells and AFP+ cells, Snap29 KO ESCs might show defective capacity to differentiate into progenitors or functional cells, especially in the ectoderm and endoderm (for detailed discussion, revised Line 430-442, 451-461).

Reviewer 2 Report

In this paper the authors have studied the role of Snap29 in the maintenance of self-renewal and differentiation as well as the its role in autophagy in an embryonic cell line. Overall, the study is well structured, however I have mayor comments:

·       In the generation of the Snap29 KO ESC you obtain two different clones one with a 28 bp deletion and the other one with a 1 bp insert. Although it is mentioned in the supplementary figure, there is no information about which clone is KO-1 and KO-2 throughout the main text. In the same way, there is no information about which clone was used for the RNA-seq analysis.

·      In figure 1, the authors conclude that the self-renewal capacity of the Snap29 KO cells is not altered. They show a colony formation assay and proliferation (MTT) assay. The MTT assay is not sufficient to proof self-renewal. It would be recommended to perform an accumulative growth curve that shows that the cells continue proliferating in the same rate as the WT controls after several passages. For how long have the authors grown the cells?

·  After paragraph 1 of results 3.1 the authors conclude that they have “successfully established the Snap29 knockout ESC lines and found that disruption of Snap29 did not influence the self-renewal maintenance of ESCs”. This conclusion should be moved and slightly changed, to include also the observation that it does not influence the maintenance of pluripotent markers, to the end of the section as they continue validating their model throughout the section.

·       Figure S2D is not included in the text. Please include it or remove the figure panel.

·     In section 3.2, how would the authors explain the larger size of the Snap29 KO EBs compared to the WT controls? The confocal images form the correct differentiation to the three germ layers (figures S3 C-E) should be included in the main manuscript.

·  In figure 6, quantifications of protein levels form the western blots experiments are missing, please include them.

·      he discussion is very poor, although it is very well discussed the autophagy results, the rest of the discussion is mostly repetition of the results obtained. How would you explain the enriched DEGs “in several pathways, including breast cancer, basal cell carcinoma, transcription misregulation in cancer, gastric cancer and others” in figure 2F? How would the authors explain the transcriptome analysis showing a change in expression of genes related to embryonic germ layer differentiation and embryonic organ development, but don´t see any differences in section 3.2. Moreover, if loss of functional Snap29 results in CEDNIK syndrome, why do the authors don´t see any changes in neural differentiation?

     Finally, I would hardly recommend to review the English, as there are several language mistakes including grammar ones that should be edited.

Author Response

Response to Reviewer 2 Comments

Thank you very much for reviewing our manuscript and providing constructive suggestions. As below, I would like to clarify the concerns point by point.

Comments and Suggestions for Authors

In this paper the authors have studied the role of Snap29 in the maintenance of self-renewal and differentiation as well as the its role in autophagy in an embryonic cell line. Overall, the study is well structured, however I have mayor comments:

  • In the generation of the Snap29 KO ESC you obtain two different clones one with a 28 bp deletion and the other one with a 1 bp insert. Although it is mentioned in the supplementary figure, there is no information about which clone is KO-1 and KO-2 throughout the main text. In the same way, there is no information about which clone was used for the RNA-seq analysis.

Response: Thank you for the comment. We have added the relevant information in the revised main text (Line 204-205, 228-230) and figure legends (Line 509-510).

  • In figure 1, the authors conclude that the self-renewal capacity of the Snap29 KO cells is not altered. They show a colony formation assay and proliferation (MTT) assay. The MTT assay is not sufficient to proof self-renewal. It would be recommended to perform an accumulative growth curve that shows that the cells continue proliferating in the same rate as the WT controls after several passages. For how long have the authors grown the cells?

Response:  Thank you for the advice.

  • An accumulative growth curve has been included in the revised Fig. 1B and the revised text (Line 210-211). Meanwhile, the MTT result has been moved into the revised Fig. S1.
  • For the majority of experiments in our study, WT and Snap29 KO ESCs at passage 4~6 after stable clonal establishment were used. Meanwhile, we grown those cells for over 20 passages and did not find the defective self-renewal of Snap29 KO ESCs.

  • After paragraph 1 of results 3.1 the authors conclude that they have “successfully established the Snap29 knockout ESC lines and found that disruption of Snap29 did not influence the self-renewal maintenance of ESCs”. This conclusion should be moved and slightly changed, to include also the observation that it does not influence the maintenance of pluripotent markers, to the end of the section as they continue validating their model throughout the section.

Response:  We appreciate the constructive suggestion and have revised the text accordingly (Line 218-220, Line 245-247).

  • Figure S2D is not included in the text. Please include it or remove the figure panel.

Response:  Fig. S2D has been included in the revised text (Line 327-329).

  • In section 3.2, how would the authors explain the larger size of the Snap29 KO EBs compared to the WT controls? The confocal images form the correct differentiation to the three germ layers (figures S3 C-E) should be included in the main manuscript.

Response: Thank you for the comment.

  • As reported, the size of EBs influences the differentiation potential of ESCs (Barzegari et al., J Cell Physiol 2020), and larger EBs are associated with enhanced cardiogenesis of ESCs (Hwang et al., Proc Natl Acad Sci U S A. 2009). Thus, we speculate that larger EBs might arise from increased differentiation of Snap29 KO ESCs into heart-beating cardiac cells. Relevant disscussion has been added in the revised text (408-414).
  • The images have been moved to the revised Fig. 3. Meanwhile, we added the relevant Western blot results detecting α-SMA, AFP, and βIII-tubulin (revised Fig.3, revised text Line 267-273). Our data indicate that the molecular marker of mesoderm, α-SMA, is upregulated in Snap29 KO differentiated cells in comparison with control cells. The results are consistent with our main findings obtained by heart-like beating and transcriptome analysis.

  • In figure 6, quantifications of protein levels form the western blots experiments are missing, please include them.

Response: Quantifications have been added in the revised Fig. 6.

  • The discussion is very poor, although it is very well discussed the autophagy results, the rest of the discussion is mostly repetition of the results obtained. How would you explain the enriched DEGs “in several pathways, including breast cancer, basal cell carcinoma, transcription misregulation in cancer, gastric cancer and others” in figure 2F? How would the authors explain the transcriptome analysis showing a change in expression of genes related to embryonic germ layer differentiation and embryonic organ development, but don´t see any differences in section 3.2. Moreover, if loss of functional Snap29 results in CEDNIK syndrome, why do the authors don´t see any changes in neural differentiation?

Response: Thank you very much for the detailed and constructive suggestion. Accordingly, we have revised the discussion section thoroughly. For detailed information, you can find in:

  • Discussion of “DEGs in Fig. 2F”: Line 356-365.
  • Newly added results and discussion regarding “transcriptome analysis and section 3.2”: revised Fig. 3, revised text Line 430-442, 452-461.
  • Discussion of “CEDNIK syndrome and neural differentiation”: Line 452-461.

     Finally, I would hardly recommend to review the English, as there are several language mistakes including grammar ones that should be edited.

Response: Thank you for the advice. We have checked the text with detail and improved the English language of our manuscript.

Round 2

Reviewer 2 Report

The authors have very much improved the manuscript.

They should add in the material and methods the methodology used for the accumulative growth curve added to Figure 1.

Author Response

Comments and Suggestions for Authors

The authors have very much improved the manuscript.

They should add in the material and methods the methodology used for the accumulative growth curve added to Figure 1.

Response: Thank you very much for providing the valuable comments. Accordingly, we have added the methodology used for the accumulative growth curve added to Fig. 1 in the revised text (Line 99-104). We hope that you will be satisfied with our  revisions.
